# Integrated Multi-Tissue Transcriptome Profiling Characterizes the Genetic Basis and Biomarkers Affecting Reproduction in Sheep (*Ovis aries*)

**DOI:** 10.3390/genes14101881

**Published:** 2023-09-27

**Authors:** Zaixia Liu, Shaoyin Fu, Xiaolong He, Lingli Dai, Xuewen Liu, Caixia Shi, Mingjuan Gu, Yu Wang, Lili Guo, Yanchun Bao, Chencheng Chang, Yongbin Liu, Wenguang Zhang

**Affiliations:** 1College of Animal Science, Inner Mongolia Agricultural University, Hohhot 010018, China; 1181703365@emails.imau.edu.cn (Z.L.); lingli20022006@sina.com (L.D.); narisu@swu.edu.cn (N.); caixiashi@imau.edu.cn (C.S.); mngmingjuangu@imau.edu.cn (M.G.); 13848126629@163.com (M.); 13474912747@163.com (L.G.); byc107054@163.com (Y.B.); baiyinbatu@163.com (B.); changchencheng8112@163.com (C.C.); 2Inner Mongolia Engineering Research Center of Genomic Big Data for Agriculture, Hohhot 010018, China; 3Institute of Animal Husbandry, Inner Mongolia Academy of Agricultural and Animal Husbandry Sciences, Hohhot 010031, China; fsy@imaaahs.ac.cn (S.F.); xl@imaaahs.ac.cn (X.H.); 4Veterinary Research Institute, Inner Mongolia Academy of Agricultural and Animal Husbandry Sciences, Hohhot 010031, China; 5Animal Husbandry and Bioengineering, College of Agronomy, Xing’an Vocational and Technical College, Ulanhot 137400, China; xamlxw8718@126.com; 6College of Veterinary Medicine, Inner Mongolia Agricultural University, Hohhot 010018, China; wangyu@imau.edu.cn; 7School of Life Science, Inner Mongolia University, Hohhot 010021, China; ybliu@imu.edu.cn; 8College of Life Science, Inner Mongolia Agricultural University, Hohhot 010018, China

**Keywords:** sheep breeding, reproductive trait, transcription atlas, HPO

## Abstract

The heritability of litter size in sheep is low and controlled by multiple genes, but the research on its related genes is not sufficient. Here, to explore the expression pattern of multi-tissue genes in Chinese native sheep, we selected 10 tissues of the three adult ewes with the highest estimated breeding value in the early study of the prolific Xinggao sheep population. The global gene expression analysis showed that the ovary, uterus, and hypothalamus expressed the most genes. Using the Uniform Manifold Approximation and Projection (UMAP) cluster analysis, these samples were clustered into eight clusters. The functional enrichment analysis showed that the genes expressed in the spleen, uterus, and ovary were significantly enriched in the Ataxia Telangiectasia Mutated Protein (ATM) signaling pathway, and most genes in the liver, spleen, and ovary were enriched in the immune response pathway. Moreover, we focus on the expression genes of the hypothalamic–pituitary–ovarian axis (HPO) and found that 11,016 genes were co-expressed in the three tissues, and different tissues have different functions, but the oxytocin signaling pathway was widely enriched. To further explore the differences in the expression genes (DEGs) of HPO in different sheep breeds, we downloaded the transcriptome data in the public data, and the analysis of DEGs (Xinggao sheep vs. Sunite sheep in Hypothalamus, Xinggao sheep vs. Sunite sheep in Pituitary, and Xinggao sheep vs. Suffolk sheep in Ovary) revealed the neuroactive ligand–receptor interactions. In addition, the gene subsets of the transcription factors (TFs) of DEGs were identified. The results suggest that 51 TF genes and the homeobox TF may play an important role in transcriptional variation across the HPO. Altogether, our study provided the first fundamental resource to investigate the physiological functions and regulation mechanisms in sheep. This important data contributes to improving our understanding of the reproductive biology of sheep and isolating effecting molecular markers that can be used for genetic selection in sheep.

## 1. Introduction

The population of sheep in China has increased from 299.3 million to 305.07 million from 2016 to 2021, with a stable number of sheep. Sheep have become an important animal model in agriculture, pharmaceutical, and biomedical research because of their suitable size, rough feeding tolerance, and short gestation period [1,2]. The increase in lambing rate can reduce the cost of ewe breeding, increase group productivity, and reduce the environmental load, which is of great significance in alleviating environmental degradation and grassland degradation. However, the domestic varieties are slow in growth, poor in reproductive performance, lagging in performance measurement, insufficient in breeding intensity, slow in progressing population genetics, and has a weak population structure. The continuous development and application of gene technology are key factors in promoting sheep production. The developed genomic tools include the 50 K SNP chip, the 600 K high-density SNP chip, and the 40 K liquid chip, as well as whole gene association analysis and joint analysis of each group, etc. We use the markers of these tools to capture the resistance/susceptibility of many diseases/conditions and to screen the various sites of interesting traits. RNA sequencing (RNA-seq) was applied to the study of several aspects of RNA biology, encompassing single-cell sequencing and translation (translatome), as well as the RNA structure (structuralome) and spatial transcriptome (spatialomics), with nearly 100 different methods derived from standard RNA-seq [3]. It is crucial to reveal the molecular composition of cells and tissues, to quantify the changes in the expression of each transcript during development and under different conditions, and to understand the development and disease [4]. In terms of animal tissues, Chen et al. [5] analyzed the ovarian tissue transcriptome of high-breeding Qira black and low-breeding Hetian sheep and screened for the DEGs involved in signaling pathways such as TGF-β, insulin, Wnt, and Notch. Shariati et al. [6] identified 5968 differentially expressed genes in the ovaries of single and multiparous sheep via RNA-seq, of which 3047 were down-regulated and 2921 were up-regulated, where the selected *AKT*3, *MAPK*8, *MAPK*9, and *RELA* genes were related to the fecundity of multiparous sheep, and most of the DEGs were involved in folliculogenesis, ovulation, and embryonic development. 

In sheep reproduction, multiple internal factors influence reproduction, such as hormones, cytokines, regulatory factors, genes, and proteins, and also multiple signaling pathways [7,8,9]. The hypothalamus is an important part of the central nervous system, sensing changes in light and dark signals from the external environment via the retina–supraoptic nucleus bundle, synchronizing the biological rhythms with changes in the environmental light and dark. The pituitary gland influences animal endocrine production, secreting a variety of hormones that are not only controlled by the hypothalamus but also act via FSH (Follicle-Stimulating Hormone) and LH (Luteinizing Hormone) negative feedback. The function of the ovary, an essential female reproductive organ, is regulated via the Hypothalamus–Pituitary–Ovary axis (HPO), which regulates the synthesis and secretion of reproductive hormones in the hypothalamus and pituitary gland via positive and negative feedback mechanisms [10]. The HPO axis mediates the direct or indirect regulation of ovulation cycles and egg production via the levels of reproductive hormones secreted by the hypothalamus, pituitary, and ovary [11,12]. The gonadotropin-releasing hormone (GnRH) stimulates the secretion of FSH and LH in the anterior pituitary and then triggers the release of E_2_ and P_4_ in ovarian follicles. Under the negative feedback effect, it inhibits the release of GnRH and FSH, thereby maintaining the regulation of reproductive hormones [13], and these hormones are secreted differently in different states of animals.

In this study, we clarified a comprehensive transcriptome analysis of multiple tissues in sheep. Our results uncovered the transcriptome atlas and analyzed gene expression as well as the transcription factors in HPO tissue. It was also demonstrated that different tissues exerted a diverse range of physiological functions, including behavior, immune response, and development. Finally, we have successfully identified the gene subsets of TFs and discussed the potential mechanism in the HPO axis. Overall, our findings offer a comprehensive analysis of gene expression variation and physiological function specialization in multiple tissues of high-fecundity sheep. It provides basic and useful resources for the further study of the molecular characteristics and transcriptional regulation of sheep.

## 2. Materials and Methods

### 2.1. Experimental Animals and Samples’ Preparation

Based on previous results on the estimated breeding value of the litter size in Xinggao sheep [14], three 3.5-year-old adult ewes with similar ages and weights were selected with higher estimated breeding values (0.32, 0.29, and 0.21). The 10 tissue samples were collected including the Hypothalamus (B), Pituitary (P), Ovary (O), Heart (H), Liver (L), Spleen (S), Kidney (K), Uterine Horn (UH), Uterine Body (UB), and Muscle (M) and quickly placed in liquid nitrogen for quick freezing. Each sheep had two replicates per tissue, a total of 60 samples. The samples were taken back to the laboratory for total RNA extraction and the collection of all samples was approved.

The transcriptome data of 16 samples were downloaded from the SRA (Sequence Read Archive) in the NCBI database (National Center for Biotechnology Information) (PRJNA597506, PRJNA600124, and PRJNA604111). Including six hypothalamus tissues of a 3-year-old female Sunite sheep, six pituitary tissue samples from a 3-year-old female Sunite sheep, and four ovary tissue samples from a 2-year-old female Suffolk sheep. 

### 2.2. RNA Extraction and Sequencing

Total RNA was extracted from 60 samples using the standard TRIzol Reagent (TaKaRa, Dalian, China). The Fragment Analyzer 5400 (Agilent) system (Agilent, Santa Clara, CA, USA) was used to ensure total RNA quantity and quality.

After the RNA samples were qualified, the eukaryotic mRNA was enriched via magnetic beads with Oligo (dT). Subsequently, a fragmentation buffer was added to break the mRNA into short fragments, and mRNA was used as a template to synthesize the one-strand cDNA with random hexamers. Then, the buffer, dNTPs, DNA polymerase I, and RNase H were added to synthesize the two-strand cDNA, and then the double-strand cDNA was purified via AMPure XP beads. The purified double-stranded cDNA was subjected to terminal repair, A tail, and the ligation of sequencing adapters, and then the fragment size was selected via AMPure XP beads. Then, PCR was performed with Phusion High-Fidelity DNA polymerase, Universal PCR primers, and the Index (X) Primer. Finally, PCR amplification was performed and the PCR products were purified with AMPure XP beads to obtain the final library. After the construction of the library was completed, Qubit 2.0 was used for preliminary quantification and the dilution of the library, and then Agilent 2100 was used to detect the size of the inserted fragments of the library. After the inserted fragments met the expectations, the effective concentration of the library was accurately quantified via the q-PCR method to ensure the quality of the library. After the library was qualified, different libraries were pooled into flow cells according to the effective concentration and the amount of the target data. Illumina’s high-throughput sequencing platform NovaSeq 6000 was used.

### 2.3. Sequencing Data Alignment and Quantitative Analysis

The transcriptome raw data were quality-controlled and filtered via Fastp software (version: 0.23.2) [15]. Removing the adaptor sequence and removing the sequence with an N ratio greater than 10% resulted in high-quality data for subsequent gene expression quantification and the construction of transcriptional profiles.

The Ovis aries reference genome (Oar_v4.0) was downloaded from the NCBI database (https://www.ncbi.nlm.nih.gov/, accessed on 5 May 2023), and HISAT2 [16] was used to construct an index, which was then used to align the clean data onto the reference genome, resulting in SAM files. The SAM files were converted into BAM files using samtools. StringTie software (version: 2.1.5) [17] was used for transcript assembly and the quantification of expression levels from the BAM files, yielding FPKM (Fragments Per Kilobase of exon model per Million mapped fragments) values for normalized transcripts and gene expression.

### 2.4. Construction of the Sheep Transcriptome Atlas

Data preprocessing: first, R packages dplyr, reshape2, and knitr were used to convert the clean data count matrix into a Cell Ranger recognizable data matrix, with the first column as the gene number, the second column as the sample number, and the third column as the corresponding count [18]. Then, the quality of the count matrix was calculated using Cell Ranger, obtaining information such as high-quality gene numbers, sequencing saturation, etc. The Seurat package was used for further data quality control, and the data was controlled according to two indicators: nCount_RNA representing the read counts for each sample, and nFeature_RNA representing the number of genes detected in each sample; the mitochondrial genes were removed. Dimension reduction and clustering analysis: 2000 highly variable genes (HVG) were screened via the FindVariableFeatures function of Seurat software (version:4.0, http://satijalab.org/seurat/) for linear change processing [19]. The principal component analysis (PCA) was performed on the gene expression level of all samples, and then the PCA results were visualized in two-dimensional space via uniform manifold approximation and projection (UMAP).

Marker gene identification of each tissue sample: marker gene identification was performed using the Seurat package’s FindAllMarkers function to find out the genes that were differentially up-regulated between each tissue group in other tissues. The VlnPlot and FeaturePlot functions are used to visualize and recognize marker genes.

### 2.5. Comparative Analysis of HPO Axis Differences in Different Breeds of Sheep

A total of thirty-two samples, collected from different breeds of sheep, were analyzed from the Hypothalamic–Pituitary–Ovarian axis (HPO) tissues. The library construction for the differential expression genes’ (DEGs) sequencing was the same as described above. Prior to the differential gene expression analysis, the batch effect of three different sources was corrected using the R package sva [20]. The PCA and cluster analysis were carried out on the data before and after removing the batch effect to show the batch effects. 

The DEG analysis was performed on the read counts for each sample after adjusting for batch effects. DESeq2 (version: 3.16) [21], which can use shrinkage estimation for dispersions and fold changes to improve the stability and interpretability of estimates, was used to identify the DEGs. The DEGs’ analysis was carried out to make the three different sources comparable to the same tissue in Hypothalamus (Xinggao sheep vs. Sunite sheep), Pituitary (Xinggao sheep vs. Sunite sheep), and Ovary (Xinggao sheep vs. Suffolk sheep). According to the results of the differential expression analysis, a *p*-value less than 0.05 and abs(log 2(fold change) > 2 are considered as the cut-offs for defining the DEGs. Between the differential breeds of sheep in the same tissue, the 1408, 1621, and 153 DEGs for the hypothalamus, pituitary, and ovary were identified via DESeq2, respectively.

Transcription factors (TFs) are thought to recognize specific DNA sequences and play vital roles in the regulation of gene expression [22]. TFs were predicted via the AnimalTFDB4.0 database (http://bioinfo.life.hust.edu.cn/AnimalTFDB4/#/, accessed on 7 June 2023) [23]. A heatmap was generated using the pheatmap R package in RStudio software (version: 4.1.2).

### 2.6. Gene Enrichment and Functional Analysis of the Hypothalamus–Pituitary–Ovary Axis

To predict the function of the hypothalamus, pituitary, and ovary samples, we performed a functional enrichment analysis of the top 3000 tissue-specific expressed genes in each tissue of Xinggao sheep. The DEGs identified among the three tissues of Sunite sheep, Suffolk sheep, and Xinggao sheep were analyzed. The online bioinformatics database Metascape (https://metascape.org/, accessed on 10 June 2023) and Cytoscape plungin ClueGO (version:3.9.1) [24] were used for the gene enrichment analysis. The enrichment items included the biological processes (BP), molecular functions (MF), and cellular components (CC) in the Gene Ontology (GO) function. The Kyoto Encyclopedia of Genes and Genomes (KEGG) was used to select entries with enrichment entries *p* < 0.01, Min Overlap = 3, Min Enrichment = 1.5. Finally, the gene set was automatically identified and annotated using the Cytoscape plug-in AutoAnnotate.

## 3. Results

### 3.1. Quality Analysis of RNA Sequencing Data

Through the sequencing of sixty cDNA libraries, the Illumina A total 410.31 Gb RNA-seq raw sequences were obtained from the 10 tissues with an average of 6.84 Gb per sample and a mean length of 150 bp. After tight filtering, a total of 2,776,050,100 clean reads (with an average of 46,267,502) were obtained. The clean reads were mapped to the reference genome with a mapped mean rate of 89.61% (Appendix A). However, the mapped rate between the P1_B344 and P2_B344 samples was low, so they were deleted, and the remaining 58 samples were used for subsequent analysis. The sixteen HPO RNA-seq samples of Sunite sheep and Suffolk sheep with clean reads were mapped to Oar_v4.0 with a mean ratio of 89.2%. After filtering, 862,427,497 clean reads (with an average of 53,901,718) were obtained.

### 3.2. Global Pattern of Gene Expression of Xinggao Sheep

A total of 25,341 genes were identified from 58 transcriptome data. To eliminate false positives, the genes with FPKM > 0.1 were screened, and the number of genes expressed in ten tissues ranged from 7129 to 15,034 (FPKM > 0.1 in all samples of the tissue). The global gene expression analysis showed that the ovary, uterus, and hypothalamus expressed the most genes, accounting for 85.5%, 87.5%, and 86.9% of all expressed genes, respectively. The hypothalamus and ovaries express a more complex transcriptome, expressing more genes (15,034 in the hypothalamus; 14,919 in the ovary), whereas 6265 genes were expressed in all tissues (Table 1). Each ubiquitous gene was expressed in all tissues in roughly the same order of magnitude.

We further explored the similarities and differences in gene expression across different tissues, as well as the enrichment of various pathways in different tissues. Specifically, the high percentage of shared gene expression between the ovaries and the uterine body and horn indicates their gene expression similarity (Figure 1). Additionally, it was found that the genes in the ATM (Ataxia Telangiectasia Mutated Protein) signaling pathway were highly significantly enriched in the spleen, ovary, uterine body, and horn (*p* < 0.001); while gene enrichment in the BMP receiving signaling pathway was highly significantly enriched in the hypothalamus, kidney, muscle, ovary, spleen, uterine body, and horn (*p* < 0.01). Furthermore, the liver, spleen, and ovaries display the highest gene enrichment in the immune response pathway, highlighting their critical role in the immune response (Figure 2).

### 3.3. Transcriptome Atlas Construction for Xinggao Sheep Tissues

The Seurat package was utilized to conduct the linear dimensionality reduction in the data by employing principal components (PCA). The genetic expression levels were sorted via the PCA values, and the genes with the highest variability in PCA 1:6 were visualized (Figure 3). To reduce technical noise in the data, the Seurat employed PCA values to cluster the genes and utilized jackstraw resampling analysis to determine the appropriate dimensions for downstream analysis.

The results of the MNN (Mutual Nearest Neighbors) dimension reduction were used. The samples were clustered and visualized via the UMAP (Uniform Manifold Approximation and Projection) nonlinear dimension reduction algorithm. The clustering algorithm used SNN (Shared Nearest Neighbor). Finally, eight clusters (Figure 4A) were obtained, and the cluster with a closer distance had a similar transcriptome. The liver and kidney were clustered into one group; the muscle, spleen, and kidney were clustered into one group; and the uterus and ovary had more similar expression patterns. It can be seen from the UMAP clustering diagram that apart from the pituitary and spleen, the cluster analysis of the top 10 genes expression in each cluster from the same cluster of genes may have similar functions from the same type of tissue. For example, *SMOC*2 is highly expressed in the uterus, and *TIMP*1, *INHA*, and *INHBA* are highly expressed in the ovary (Figure 4B).

### 3.4. Transcriptome Analysis of Hypothalamus–Pituitary–Ovarian Axis in Xinggao Sheep

An initial overview of the gene expression profiles of 16 high-quality HPO samples was explored. Through sequencing, a total of 114.41 Gb RNA-seq raw sequences were obtained from the HPO axis with an average of 7.15 Gb per sample and a mean length of 150 bp. After the filter, a total of 749,546,610 clean reads were obtained; an average of 46,846,663 reads per sample. The clean reads were mapped to the reference genome with the mapped rate ranging from 86.68% to 93.65%. To explore the expressed genes between the hypothalamus, pituitary, and ovary tissues, the gene expression levels were quantified via FPKM, and a total of 23,747 genes were identified in all samples. The 8035 genes at 0.1 < FPKM ≤ 5 accounted for 33.8% of all expressed genes; 7525 genes in 5 < FPKM ≤ 100 accounted for 31.7% of all expressed genes; and 631 genes > 100 accounted for 2.66% of all expressed genes (Figure 5). The genes expressed in each tissue were screened according to FPKM ≥ 0.1, where low-expressed genes were excluded, and the results showed that a total of 15,035 genes were expressed in the hypothalamus, 11,446 genes in the pituitary, 14,919 genes in the ovary, 11,016 genes in all tissues, 2871 genes co-expressed in only two tissues, and 2609 genes expressed in only one (Figure 6).

To elucidate the function of the genes expressed in the HPO axis, gene ontology (GO) and KEGG enrichment analyses were performed on the top 3000 expressed genes in each tissue. The results showed that these genes were mainly enriched in various biological processes, cellular components, molecular functions, and pathways, including the mitochondrial matrix, translation, ribosome biogenesis, postsynapse, and protein folding (Figure 7). However, there were differences in the enriched categories among the hypothalamus, pituitary, and ovary. Specifically, in the hypothalamus, the 478 BP terms, 89 CC terms, 96 MF terms, and 25 KEGG pathways. BP was enriched in the regulation of trans-synaptic signaling, the modulation of chemical synaptic, telencephalon development, the regulation of hormone secretion, etc.; CC was enriched in the microtubule-associated complex, the neuromuscular junction, and so on; and MF was enriched in the monoatomic ion transmembrane transporter activity, kinesin binding, etc. The KEGG pathway analysis revealed that the top enriched pathways include circadian entrainment, insulin secretion, and the PPAR signaling pathway.

In the pituitary tissue, the top expressed genes were significantly enriched in 247 terms, including 162 BP terms, 31 CC terms, 47 MF terms, and 7 KEGG pathways. The BP terms were predominantly related to vesicle fusion with the Golgi apparatus and the negative regulation of histone modification, while the CC terms were primarily associated with the DNA-directed RNA polymerase complex, RNA polymerase complex, ATPase regulator activity, and RNA polymerase II complex binding. The enriched KEGG pathways were the basal transcription factors, glycerolipid metabolism, purine metabolism, and arginine biosynthesis.

In the ovary, the top expressed genes were enriched in 665 terms, including 538 BP terms, 40 CC terms, 65 MF terms, and 22 KEGG pathways. The BP terms were mainly related to reproductive system development, embryonic morphogenesis, and reproductive structure development. The enriched KEGG pathways were ECM–receptor interaction, the p53 signaling pathway, and the Wnt signaling pathway. Notably, the oxytocin signaling pathway was simultaneously detected in all three tissues.

To further determine the roles of specific signaling pathways in sheep reproductive regulation, GO and KEGG pathway enrichment analyses were performed on the tissue-specific expressed genes in the three tissues. Figure 8 shows the top 20 enriched pathways in each tissue. It was found that, except for the biological process involved in interspecies interaction between the organisms and viral processes, all other top enriched items in the hypothalamus were significant, indicating the importance of the hypothalamus in life activities. The ovary was mainly significantly enriched in the developmental process, regulation of biological processes, immune system process, and reproductive process. The pituitary was mainly enriched in localization, signaling, the multicellular organismal process, and the developmental process, with the developmental process showing significant shared enrichment in the hypothalamus, pituitary, and ovary tissues.

### 3.5. Differential Expression Analysis of HPO Axis Tissues in Different Breeds of Sheep

The numbers of expressed different genes in the samples from Sunite sheep and Suffolk sheep were 22,874, among which the reads’ count >10 in at least two samples of the HPO tissue via the removing batch effect, with different data sources clustered onto different branches. Figure 9 illustrates that each tissue was divided into two groups based on the principal component analysis, and most of the samples in each group came from the same source, indicating heterogeneity in the data. After correcting for the batch effect via sva, the batch effect, tissue effect, and breed effect were all reduced. PC1 hypothalamus, pituitary, and ovarian tissues decreased from 53.5 to 39.5%, 44.7% to 29.6%, and 37.9% to 35.4%, respectively.

We conducted the differential gene expression analysis of all 22,874 genes between the differential breeds of sheep in the same tissue. After the removal of the batch effect, there were 2661 DEGs, of which 1408, 1621, and 153 DEGs for the hypothalamus, pituitary, and ovary were identified via DESeq2, respectively. The Gene Ontology analysis revealed that the DEGs were enriched in functions for the neuroactive ligand–receptor interaction, sensory perception of light stimulus, receptor–ligand activity, and so on, which, through mutual regulation, mutual restriction, and mutual influence, the female’s endocrine was regulated to maintain the homeostasis of the internal and external environment and material metabolism (Figure 10).

There are 1028 TF genes in the sheep genome, of which 2661 DEGs in the three breeds of sheep were identified as 51 TF genes in the HPO. There are 17 TF families and six cofactor genes (*ACTB*, *CSRP*3, *SMYD*1, *WNT*4, *TNNI*2, and *NLRP*12) in the HPO TFs. Approximately thirty percent of 51 TFs were predicted to have a homeobox motif (Figure 11). Integrating the results of the three tissues, we hypothesize that the hypothalamus and pituitary activate various functions and pathways, such as behavior, development, and the cAMP signaling pathway, then the signals transmit to the ovary and further activate gamete generation and embryo organ morphogenesis (Figure 12).

## 4. Discussion

Gene regulatory networks, as continuous and complex dynamic systems, vary with the environment. RNA-Seq can reveal the gene regulatory network of animal organisms. In the study, a large number of commonly expressed genes were found in all tissues, and these genes accounted for the majority of the mRNA library. The relative expression levels of ubiquitously expressed genes still varied among tissues. Almost all genes were expressed in the heart, liver, and spleen, as well as most transcripts came from a few highly expressed genes, while the hypothalamus, kidney, and ovary expressed more genes. Previous studies have shown that mRNAs expressed in the hypothalamus of mice and humans have abnormally long 3′ UTRs, and the average 3′ UTR length of the morphogenesis and signal transduction gene is higher, indicating that UTR-based regulation increases the complexity of genes, such as the immune response (high in lymph nodes), muscle contraction, heart development, and electron transfer (high in the heart), and the signal transduction and G protein-coupled receptor signaling (high in the hypothalamus) [25].

The ovary shared 87% of commonly expressed genes with the uterine horn and uterine body, and 97% of genes were co-expressed in the uterine horn and uterine body. Furthermore, in the transcriptional atlas, the uterine horn and uterine body tissues were clustered together and closed to the ovary. In the study, the ovary was assigned to Cluster 5, and its highly expressed genes, such as *TIMP*1, *INHA,* and *INHBA*, were predominantly localized in the ovary. Studies have shown that *TIMP*1 was located in oocytes with variable intensity; *MMP*1, *TIMP*2, and *TIMP*3 were also detected in ovarian stroma and located in cells and stroma that control a series of cell functions in specific cells and tissues [26]. The average FPKM of the *INHA* gene in the ovary is 371, which is distributed in the peripheral region of the cytoplasm in the oocytes of large follicles and exists in the whole cytoplasm in the oocytes of small follicles; after in vitro fertilization, *INHA* is strongly expressed in the oocytes of small follicles, especially in the zona pellucida. Before and after in vitro fertilization, the *INHB* protein is highly expressed in the oocytes and cytoplasm of small follicles, and INHs can be used as a marker of porcine oocyte quality [27]. *INHBA* was involved in the synthesis of inhibin A/AB. The expression of *INHBA* increases with the increasing follicle diameter and is widely distributed in ovarian tissue [28]. *INHBA* transfection affected granulosa cell proliferation and apoptosis and regulated the expression of many cell cycle and apoptosis-related genes. *INHBA* overexpression significantly reduces the secretion of activin and estradiol and increases the secretion of inhibin and progesterone. The expression of the FSHR-β subunit significantly decreases and increases with the overexpression and knockdown of *INHBA*, respectively, where the *INHBA* gene was related to follicular development by regulating proteins [29]. Meanwhile, *INHA* and *INHBB*, as regulators of hormone secretion and other follicular events, were important factors in regulating the bovine granulosa cell function [30]. The highly expressed gene *SMOC*2 (SPARC-related modular calcium binding 2) in the uterus affects growth factor signaling, migration, cell proliferation, and angiogenesis [31]. Thomas et al. [32] found that SMOC has been shown to inhibit the BMP signaling pathway downstream of the BMPR1B receptor during African clawed frog development, and loss-of-function mutations led to neural development stagnation. *SMOC*1 is specifically expressed in supporting cell progenitor cells (pre-supporting cells and pre-granulosa cells), downregulated in E13.5 ovaries in an RA-independent manner, and expressed at lower levels in nuclear receptor subfamilies and zinc-finger protein mutants [33].

Sequencing the transcriptome of tissues or cells under various conditions can identify the network pathways of gene regulation, giving insight into the changes in gene expression levels in the ovine hypothalamus, pituitary, and ovarian tissues. The regulation of energy metabolism is crucial for oogenesis, and there is a complex interaction between the genes involved in protein synthesis and storage [34]. This physiological phenomenon depends on various organs involved in reproduction. The hypothalamic–pituitary–ovarian axis plays a vital role in regulating reproduction and endocrine function in sheep. The axis is responsible for releasing LH and FSH hormones by secreting GnRH-I from the hypothalamus. These gonadotropins act on the gonads to stimulate gametogenesis in the left ovarian specialized cells and the secretion of sex steroid hormones, which play a crucial role in regulating oocyte development, follicular maturation, and ovulation by releasing sex hormones [35]. At present, there have been identified several crucial genes and signaling pathways that impact sheep reproduction, located in both the pituitary [36] and ovary [37]. Tian et al. [38] analyzed the transcriptome of ovary, pituitary, and hypothalamus tissues from twin and singleton ewes with the FecB genotype (B+ and ++) and observed that 128, 263, and 139 differentially expressed genes had the FecBB+/FecB++ genotype, indicating a synchronous and differential increase in the mRNA transcription levels of ovarian steroid synthesis signaling genes, the hypothalamic TGF-β signaling pathway, the cAMP signaling pathway, and the dopamine synaptic pathway. 

In this study, the transcriptome analysis of the hypothalamus–pituitary–ovary axis revealed several important core genes and signaling pathways regulating reproduction. The 15,035, 11,446, and 14,919 genes were expressed in the hypothalamus, pituitary, and ovary, respectively. And, oxytocin signaling has been identified as a common signaling pathway in HPO axis tissues. Since 1930, nearly 25,000 papers have been published on many aspects of the hypothalamus and peripheral oxytocin system (OXT), revealing the central role of OXT and its receptor (OXTR) in reproductive, social, and affective behavior [39]. OXTR is a 7-transmembrane G-protein-coupled receptor that binds either Gαi or Gαq proteins. It activates a series of signaling cascades such as the MAPK, PKC, PLC, or CaMK pathways that target transcription factors such as CREB or MEF-2 to regulate neurite outgrowth, cell viability, and increased survival, and OXT is also synthesized in peripheral tissues, such as the uterus, placenta, amnion, corpus luteum, testis, and heart [40]. A previous study demonstrates the specificity of the OXTR receptor and the effect of OXTR deficiency on glial cell proliferation and survival by establishing an OXTR knockdown cell line, which found that OXT can stimulate cell proliferation, increase ERK1/2 phosphorylation, promote cell proliferation, and significantly inhibit H_2_O_2_-induced ROS growth. It also prevented the H_2_O_2_- and camptothecin-induced decrease in cell viability, suggesting that OXT stimulates the proliferation of astrocyte-like cells [41]. It was observed that ovarian-expressed genes are not only enriched in reproduction-related functions, but also involved in immune system processes, and the interaction between the immune system and reproduction is multiple. Immune cells are directly or indirectly involved in the regulation of all levels of the hypothalamic–pituitary–ovarian axis through their products, and they are present in the ovary and their number increases during the cycle. During follicular development, cytokines assist granulosa cell growth while inhibiting their differentiation. Cytokines are members of a larger regulatory network present in the ovary involving hormones and growth factors [42]. 

To investigate the spatial expression changes of the hypothalamus–pituitary–ovary gonadal axis at the transcriptome level, we conducted a comparative transcriptome analysis of these three tissues in various sheep breeds. Our study revealed transcriptional differences and distinct physiological functions among these tissues. Previous research has extensively examined the differences in the HPO gonadal axis in mammals and poultry, identifying the significant involvement of nerve ligand–receptor interaction and the calcium signaling pathway in animal reproduction [43]. The results were consistent with the results of this study. Mishra et al. [44] found that 414, 356, and 10 DEGs were identified in the pituitary, ovary, and hypothalamus between high egg production and low egg production of Chinese Luhua chicken, and the functions of DEGs in different tissues were also different. 

Transcription factors (TFs) serve as key regulators and selection genes in animal growth, development, and defense response against adversity. They control the determination of cell types, development modes, and specific pathway control, such as immune response [45]. To further investigate transcriptional variation across the HPO gonadal axis, TFs were identified based on HPO’s differential expressed genes. It was found that there were 1028 TF genes in the sheep genome, of which 51 TF genes were found in 2661 DEGs. The animal transcription database showed that there were 1659, 1611, 1455, and 1442 transcription factors in human, mouse, goat, and dog, respectively [24]. Our study suggested that a majority of homeobox TFs (*HOX* genes) were differentially expressed in HPO. The homeobox TFs have been recognized as key determinants in the specification of cell fates during development [46]. In humans and other mammals, 39 *HOX* genes are clustered in four complexes called HOXA, B, C, and D [47]. A large number of studies have shown that the multifunctional homeobox-containing (*HOX*) D3 gene is involved in various physiological and pathological processes. The specific downregulation of HOXD3 resulted in the aggregation of cells in the G2 phase of the cell cycle [48]. HOXD3 also mediates novel crosstalk between endothelial BMP and TGFβ signaling, suggesting that the regulation of HOXD3 to control dysfunctional BMP and TGFβ signaling provides a possible therapeutic approach, and the cascade of BMP9-HOXD3-TGFβ can affect Notch signaling and angiogenesis [49]. 

Collectively, our comparative transcriptome analysis revealed gene expression differences in different breeds of sheep, particularly in differentially expressed transcription factor genes. This study contributes to the understanding of the functional regulation of the hypothalamus–pituitary–ovary axis in sheep. It is hypothesized that the hypothalamus and pituitary activate multiple functions and pathways, including behavior, development, and the cAMP signaling pathways, and transmit these signals to the ovary, thereby initiating gamete production and the morphogenesis of embryonic organs. To the best of our knowledge, this is the first. Although these observations are still in the primary stage, they will provide basic information for the physiological and biochemical studies of sheep reproduction and development.

## 5. Conclusions

In summary, this study aimed to explore the transcriptome atlas and gonad axis of prolific sheep and attempted to elaborate on the genetic mechanism of litter size. The study detected the most expressed genes in the hypothalamus and ovary, with the most complex transcript levels. The high-expression genes were mainly involved in the mitochondrial matrix, translation, and ribosome biogenesis. The 51 TF genes may play an important role in HPO. The goal was to lay the foundation for marker-assisted management of sheep breeding and provide important reference data for exploring the genetic basis of fecundity and improving the reproductive performance in sheep breeding.

## Figures and Tables

**Figure 1 genes-14-01881-f001:**
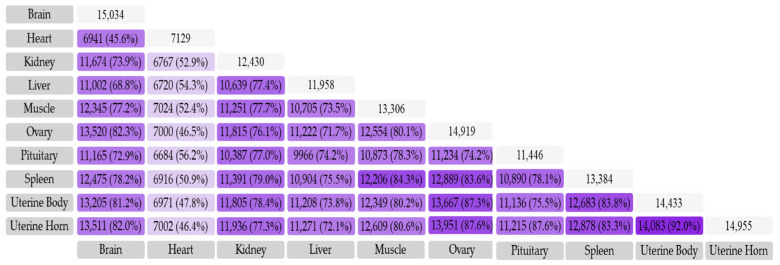
Number and proportion of commonly expressed genes in different tissues.

**Figure 2 genes-14-01881-f002:**
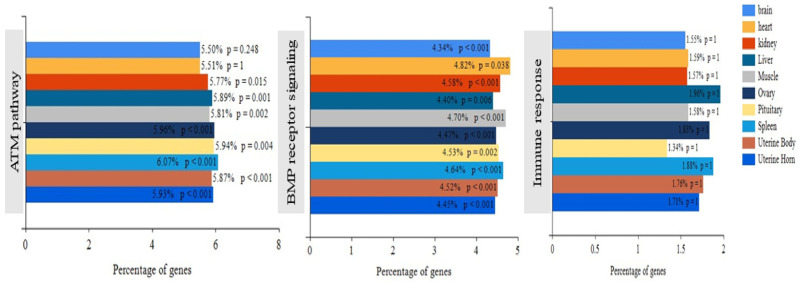
The enrichment of signaling pathways in different tissues.

**Figure 3 genes-14-01881-f003:**
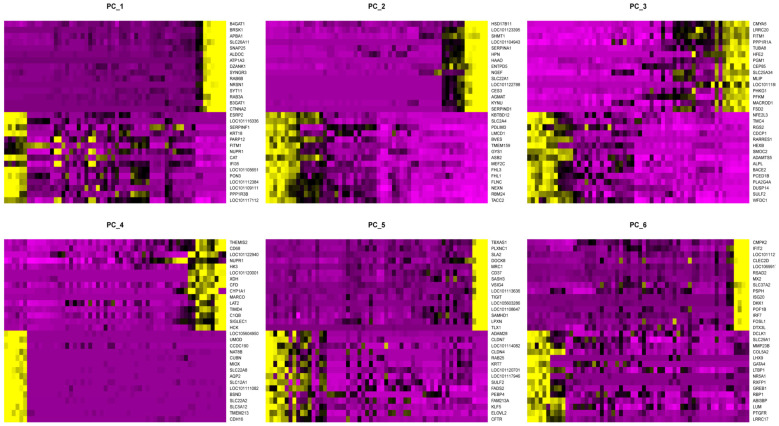
PC 1–6 High variation gene expression heatmap.

**Figure 4 genes-14-01881-f004:**
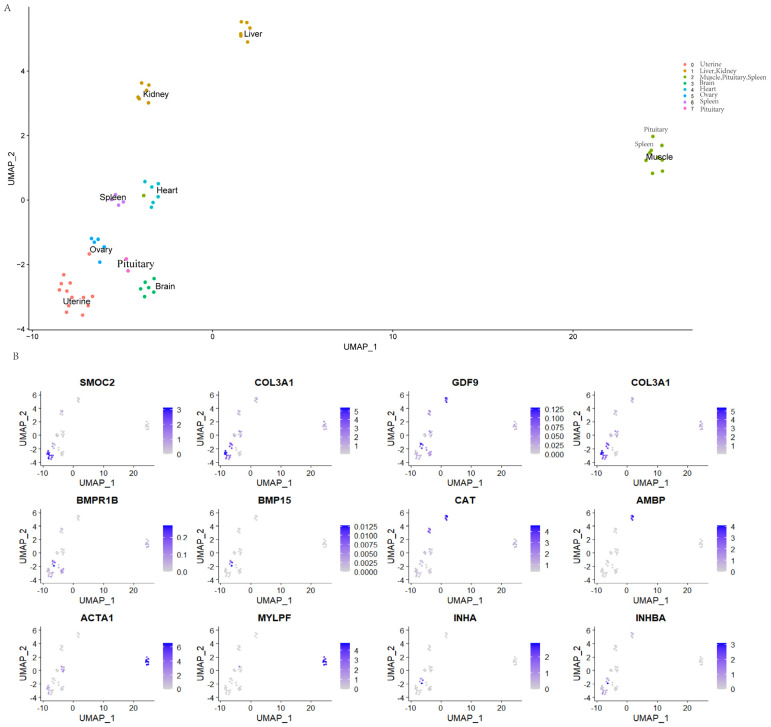
Transcriptome UMAP clustering results of 10 sheep tissues. (**A**) Transcriptome atlas, and (**B**) mapping of key genes in the tissues’ expression pattern of top gene.

**Figure 5 genes-14-01881-f005:**
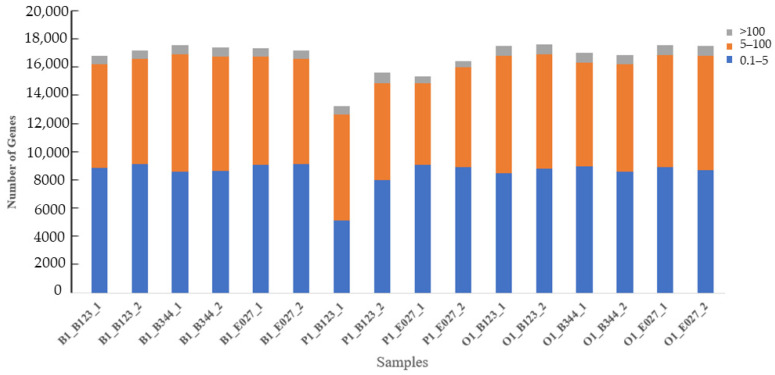
Distribution of gene expression in HPO.

**Figure 6 genes-14-01881-f006:**
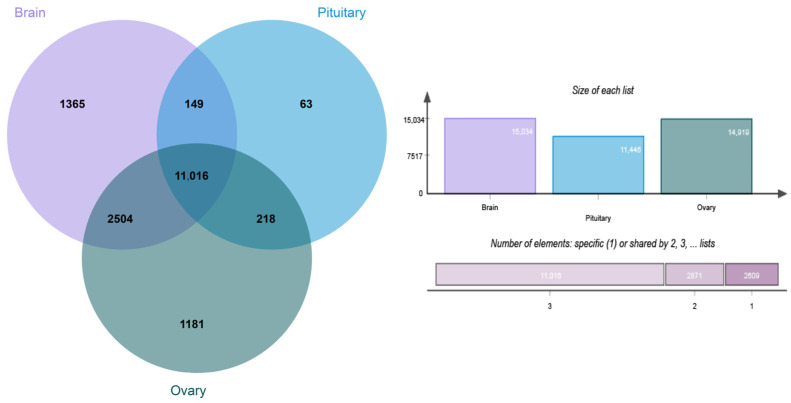
Venn map of all expressed genes under different groups.

**Figure 7 genes-14-01881-f007:**
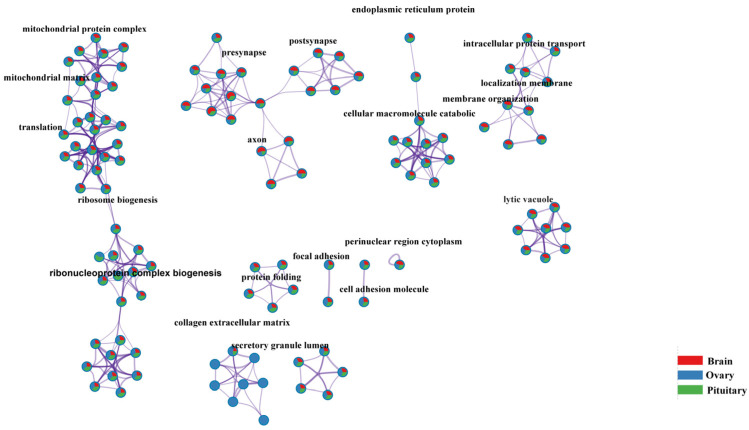
Enrichment network visualization for results from the three gene lists. Note: Nodes are represented by pie charts indicating their associations with each input study. Cluster labels were manually added. Color code represents the identities of gene lists.

**Figure 8 genes-14-01881-f008:**
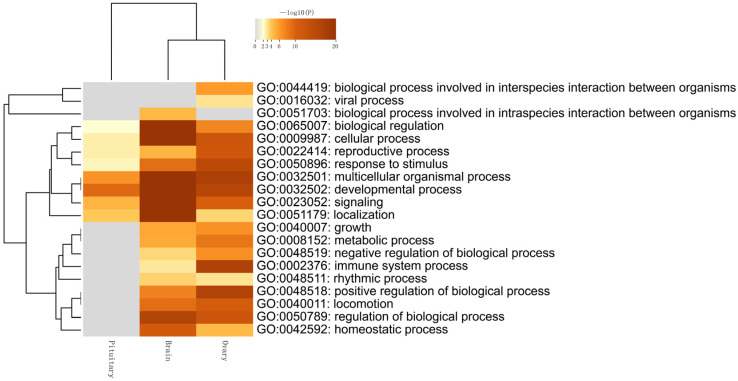
A heatmap showing the top enrichment clusters. Note: one row per cluster, using a discrete color scale to represent statistical significance; gray color indicates a lack of significance.

**Figure 9 genes-14-01881-f009:**
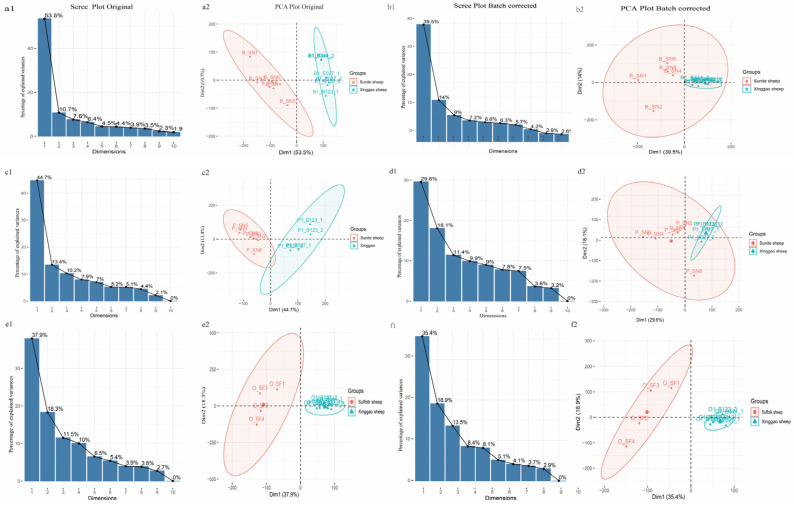
PCA before and after removing the batch effect. (**a1**,**b1**) represent the data scree of the hypothalamus before and after the batch effect was removed, respectively; (**a2**,**b2**) represent the PCA clustering diagram of the hypothalamus before and after the batch effect was removed, respectively; (**c1**,**d1**) represent the data scree of the pituitary before and after the batch effect was removed, respectively; (**c2**,**d2**) represent the PCA clustering diagram of the pituitary before and after the batch effect was removed, respectively; (**e1**,**f1**) represent the data scree of the ovary before and after the batch effect was removed, respectively; (**e2**,**f2**) represent the PCA clustering diagram of the ovary before and after the batch effect was removed, respectively.

**Figure 10 genes-14-01881-f010:**
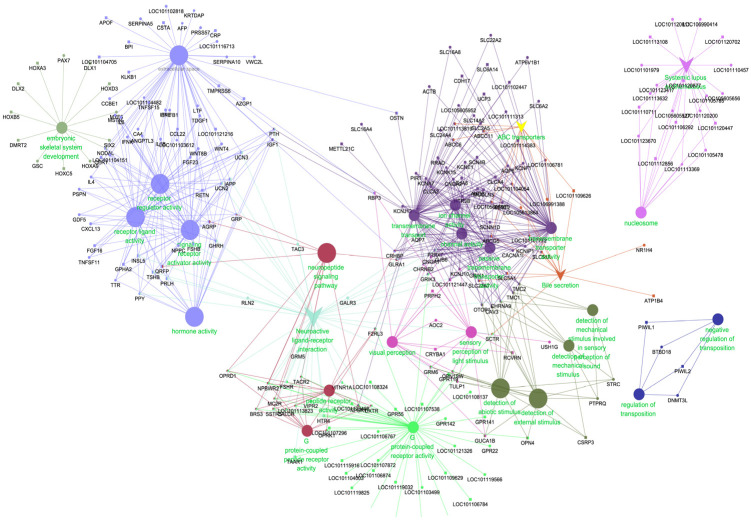
The overall analysis of differentially expressed genes.

**Figure 11 genes-14-01881-f011:**
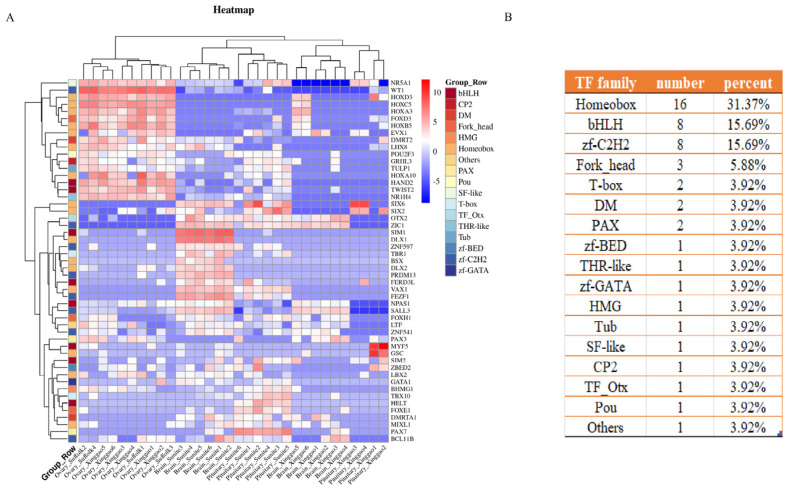
Identification of TF genes. (**A**) Heatmap analysis of DEGs’ TF genes in HPO, and (**B**) summary of TF genes in different families.

**Figure 12 genes-14-01881-f012:**
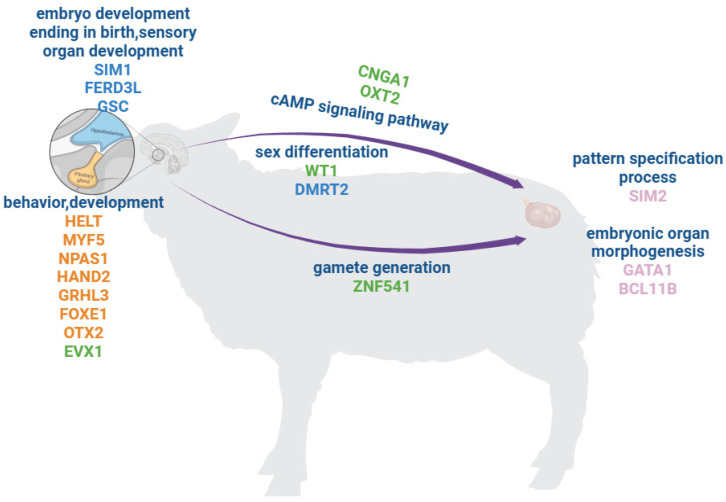
The mechanism of sheep in Hypothalamic–Pituitary–Ovarian Axis. (This image was drawn by using the online drawing software BioRender. Available at: https://app.Biorender.com/ (accessed on 2 July 2023).

**Table 1 genes-14-01881-t001:** The number of genes expressed in different groups and the number of genes with FPKM threshold.

	FPKM	>0.01	>0.1	>1	>5	>10	>100
Tissue							
Hypothalamus		16,030	15,034	11,611	6641	4271	388
Heart		8560	7129	2558	708	385	66
Kidney		12,794	12,430	9486	4900	2997	318
Liver		12,595	11,958	8059	3736	2306	275
Muscle		14,638	13,306	9588	5030	3148	419
Ovary		16,130	14,919	11,278	6761	4366	426
Pituitary		12,033	11,446	8619	4503	2728	243
Spleen		14,486	13,384	10,016	5312	3115	250
Uterine Body		15,636	14,433	10,841	5744	4266	466
Uterine Horn		16,534	14,955	11,307	6744	4442	522
On average per tissue		13,943	12,899	9336	5007	3202	337
In all 58 samples		7560	6265	2142	551	302	54

## Data Availability

The data presented in this study are available from the corresponding author upon request. The data are not available to the public due to an ongoing study for other purposes, as stipulated in a confidentiality agreement.

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
