# Peer review of "Integrated Multi-Tissue Transcriptome Profiling Characterizes the Genetic Basis and Biomarkers Affecting Reproduction in Sheep (Ovis aries)"

_genes, 2023, doi:10.3390/genes14101881_

Round 1

Reviewer 1 Report

The manuscript: "Integrated multi-tissue transcriptome profiling characterizes the genetic basis and biomarkers affecting reproduction in sheep (Ovis aries)" presents the results of the comparison of transcriptomes of several tissues (brain, pituitary, ovary, liver, spleen, heart, kidney, uterine horn, uterine body and muscles) in Xing-gao sheep. In addition, the authors compared hypothalamic, pituitary and ovarian transcriptomes from Xing-gao, Sunite, Suffolk sheep. Although the work provides a lot of new data on the ovine transcriptome, it lacks a clear research hypothesis. It is not known on what basis the authors present the potential mechanism of action of the hypothalamic-pituitary-ovarian axis. The description of the concept contains only a few sentences and it is not known which data (results of transcription factor analysis, race comparisons or tissue comparisons) were considered by the authors as priority in creating the scheme (Fig.12)

The work is written in a very chaotic way and lacks a lot of important information

More specific comments:

line 37-38 differentially expressed genes - between which samples?

line 53-55 - the sentence is repeated

line 60 - what do you mean by population sub-structure?

line 99-please explain GnRH

line 104 -please rewrite the sentence - it is unclear

lines 107-108 gonadal axis?

lines 104-108 this paragraph should contain the hypothesis and the aim of the study

line 119-122 please provide tyhe desription of the samples (breed, age, sex) and cite the source paper

lines 130-140what about indexed primers for library construction?

line 340 - DEGs for which comparison ? (You have written earlier that each breed was compared which each)

Supplementary table with DEGS from each comparison should be provided 

The RNA-seq data shoud be deposited to GEO database

Please explain how was the gene ontology analysis performed? which dataset was used for each analysis?

line 354 which DEGs? (between which breeds or tissues?

How was the hypothalamus extracted from the brain ? How can you be sure that the proper region was extracted?

line 374-375 there is not enough data to conclude that the the liver transcriptome is less complicated that ovary transcriptome, please rewrite the sentence

line 477 in which comparison?

line 496 please rewrite the sentence

Which part of the brain was used to between tissue comparison?

Author Response

Question1: line 37-38 differentially expressed genes - between which samples?

Response1: We have added the last part of the abstract, please check it at L37~L38.

To further explore the differences in the expression genes (DEGs) of HPO in different sheep breeds, we downloaded the transcriptome data of the hypothalamus, ovary, and pituitary tissues in the public data, and the analysis of DEGs (Xinggao sheep vs. Sunite sheep in Hypothalamus, Xinggao sheep vs. Sunite sheep in Pituitary, and Xinggao sheep vs. Suffolk sheep in Ovary) revealed that the neuroactive ligand-receptor interactions and receptor-ligand activity.

Question2: line 53-55 - the sentence is repeated

Response2: I am sorry for my careless writing, I have deleted it.

Question3: line 60 - what do you mean by population sub-structure?

Response3: I carefully read this sentence, and think it is not appropriate to add a sub-structure, so I have replaced sub-structure by structure.

Question4: line 99-please explain GnRH

Response4: I 've added GnRH to the manuscript in full spelling. Gonadotropin-releasing hormone (GnRH)

Question5: line 104 -please rewrite the sentence - it is unclear

Response5: This sentence have been rewritten, adding the purpose of the study.

In this study, we clarified the expression patterns of multiple tissue genes in high-breeding Xinggao sheep, and further comprehensively analyzed the HPO tissues genes and transcription factor. This study aimed to explore the atlas and gonad axis transcriptome and attempted to elaborate the genetic mechanism to further benefit local sheep breeding.

Question 6: lines 107-108 gonadal axis?

Response6: The whole sentence is modified.

Question 7: lines 104-108 this paragraph should contain the hypothesis and the aim of the study

Response 7: The whole sentence is modified

Question 8: line 119-122 please provide tyhe desription of the samples (breed, age, sex) and cite the source paper.

Response 8: We have added the these information, please check it at L133~L137.

Question 9: lines 130-140what about indexed primers for library construction?

Response 9: The PCR was performed with Phusion High-Fidelity DNA polymerase, Universal PCR primers and Index (X) Primer.

Question 10: line 340 - DEGs for which comparison ? (You have written earlier that each breed was compared which each)

Response 10: Thank you for your suggestions. I have added specific information on differentially expressed genes at Material and Method 2.5.

I have modified it to: DEGs analysis were carried out to make the three different sources comparable same tissue Hypothalamus (Xinggao sheep vs. Sunite sheep), Pituitary (Xinggao sheep vs. Sunite sheep), and Ovary (Xinggao sheep vs. Suffolk sheep). According to the results of differential expression analysis, P-value less than 0.05 and abs(log 2(fold change) > 2 as cut-offs for defining DEGs. Between differential breeds of sheep in the same tissue,the 1 408, 1 621 and 153 DEGs for the hypothalamus, pituitary, and ovary were identified by DESeq2, respectively.

Question 11: Supplementary table with DEGS from each comparison should be provided Response 11: Thank you for your suggestion. I have uploaded the table of DEGs to the supplementary.

Question 12: The RNA-seq data shoud be deposited to GEO database

Response 12: Thank you for your suggestions. I have deposited RNA-seq data to GEO database.

Question 13: Please explain how was the gene ontology analysis performed? which dataset was used for each analysis?

Response 13: Thank you for your suggestions. I revised it in the manuscript.

To predit the function of hypothalamus, pituitary and ovary samples, we per-formed functional enrichment analysis of the top 3 000 expressed genes and tissue-specific in each tissue of Xinggao sheep. And the DEGs identified among the three tis-sues of Sunite sheep, Suffolk sheep and Xinggao sheep were analyzed.

Question 14: line 354 which DEGs? (between which breeds or tissues?

Response 14: The specific grouping of DEGs have been added in method 2.5, and 2661 DEGs represent the union of differentially expressed genes in Hypothalamus (Xinggao sheep vs. Sunite sheep), Pituitary (Xinggao sheep vs. Sunite sheep), and Ovary (Xinggao sheep vs. Suffolk sheep).

Question 15: How was the hypothalamus extracted from the brain ? How can you be sure that the proper region was extracted?

Response 15: The head of the slaughtered Xinggao sheep was cut transversely at 2-4cm from the occipital condyle in front of the ear with an electric saw in the frontal bone position perpendicular to the direction of the sheep 's nose. The bone was cut into the bone and the brain tissue in the head could be seen. 2 ) The bones on both sides were split along the split, and the split part was separated to directly expose the brain tissue ; 3 ) Use ophthalmic tweezers to open the cerebellum, expose the junction of the brain stem and the brain, and then use the ophthalmic scissors to cut off a round structure along the root, which is the hypothalamus.

In this study, in order not to confuse the abbreviations in the sample, such as the abbreviation of the heart is H, I abbreviated the hypothalamus as the capital letter B. All the brain words in this study refer to the hypothalamus. In the manuscript, I have replaced the brain by hypothalamus.

Question 16: line 374-375 there is not enough data to conclude that the the liver transcriptome is less complicated that ovary transcriptome, please rewrite the sentence

Response 16: I have modified it to: Almost all genes were expressed in the heart, liver and spleen,and most transcripts came from a few highly expressed genes, while the hypothalamus, kidney, and ovary express more genes.

Question 17: line 477 in which comparison?

Response 17: The source of the differential genes has been written in Method 2.5 and Result 3.5. I hvae changed this sentence to: Found that there were 1 028 TF genes in the sheep genome, of which 51 TF genes were found in 2 661 DEGs.

Question 18: line 496 please rewrite the sentence

Response 18: Thank you for your suggestions. I revised it in the manuscript. Collectively, our comparative transcriptome analysis revealed gene expression difference in different breeds of sheep, particularly in differentially expressed transcription factor genes. This study contributes to the understanding for the functional regulation of the hypothalamus-pituitary-ovary axis in sheep. It is hypothesized that the hypothalamus and pituitary activate multiple functions and pathways, including behavior, development, and cAMP signaling pathways, and transmit these signals to the ovary, thereby initiating gamete production and morphogenesis of embryonic organs. To the best of our knowledge, this is the first. Although these observations are still in the primary stage, they will provide basic information for the physiological and biochemical studies of sheep reproduction and development.

Question 19: Which part of the brain was used to between tissue comparison?

Response 19: In this study, in order not to confuse the abbreviations in the sample, such as the abbreviation of the heart is H, I abbreviated the hypothalamus as the capital letter B. All the brain words in this study refer to the hypothalamus. In the manuscript, I have replaced the brain by hypothalamus.

Reviewer 2 Report

General comments

This study entitled “Integrated multi-tissue transcriptome profiling characterizes the genetic basis and biomarkers affecting reproduction in sheep (Ovis aries)” aimed to investigate the transcription profiles and gene expression on several tissues in Xinggao sheeps with higher estimated breeding values. This is an original research to better understand the genetic mechanisms exploring the expression pattern of multi-tissue genes mainly related with prolificacy.  The study is very well designed. Despite the fact that only 3 ewes were used, the samples were collected from 10 different types of tissues (58 samples at total, considering two replicates per tissue) to identify a large number of genes. The use of this large collection of tissues to evaluate the gene expression pattern is highly appreciated; and gives a strong scientific soundness to this work. The distinct sections, including the figures, were clearly presented contributing for the readability of the paper, and supported by adequate literature. The introduction is enougth to contextualize the nature of the study. M&M ensures the reproductibility of the experiment. Results are full described which are supported by high quality and informative figures. The discussion is appropriate regarding the findings of this work and the current literature. The conclusions are full suported by the results. This paper certainly contributes to determine the genetic mechanisms to improve local sheep breeds at reproductive level, focusing the the gene expression involved in the hypothalamic-pituitary-ovarian axis. Moreover, it can be used to elucidate the mentioned mechanisms in other breeds and identify specific biomarkers to improve the fertility in sheep through genetic selection. L111: “Based on our…”. Congratulations to the authors.

Author Response

Thank you for your affirmation of my manuscript. This gives me more confidence to do better research.

Reviewer 3 Report

This is an interesting manuscript aimed to study the transcritptome profile for several tissues associated with the reproductive trait “litter size” in sheep. However, I suggest considering some general comments, as well as several grammar details.

General comments:

-       Abstract section should be 200 words maximum according to “Instructions for Authors”.

-       Introduction section should be written in short paragraphs instead of one very long paragraph.

-       In Materials and Methods section, I suggest to include a subsection related to “Gene enrichment or functional analysis”. Also, the website address and the accession date are required for all bioinformatics resources that were used in the study.

-       In Discussion section there some very long paragraphs. I suggest separating each of them in 2-3 shorter paragraphs

-       Conclusions section appears to be a brief summary of the results. Instead, I suggest including 2 or 3 conclusive sentences describing the relevance of the study or the importance of the findings. I recommend mentioning litter size, as it was the only reproductive trait involved in this study.

-       In References section, only the first letter of the article title should be capitalized.

Grammar suggestions:

-       Line 3: I suggest to use “litter size” in the title instead of “fertility”, because it was the only reproductive trait analyzed in the study.

-       Line 4: Replace “Ovis aries” by “Ovis aries”.

-       Line 20: Replace “The heritability of sheep of litter size is low” by “The heritability of litter size in sheep is low”.

-       Lines 26 and 31: I suggest defining UMAP and ATM.

-       Line 50: I suggest replacing “stock” by “population”.

-       Line 51: Replace “Sheep have become important model animals in agriculture” by “Sheep have become an important animal model in agriculture”.

-       Lines 53-54: This sentence appeared to be repeated.

-       Line 65: Replace “The development of” by “The developed”.

-       Line 72: Separate the round bracket from the text.

-       Line 87: Replace “secretion” by “production”.

-       Line 102: Replace “homeostasis” by “regulation”.

-       Line 106: Please complete the sentence as follows “ewes with high estimated breeding value for litter size…”.

-       Line 111: Replace “Based on my previous” by “Based on previous”.

-       Line 112: Separate the square bracket from the text.

-       Line 117: Remove the dot.

-       Line 145: Separate the square bracket from the text.

-       Line 148: Replace “ovis ories” by “Ovis aries”.

-       Lines 179-181: Please remove blank spaces within the round brackets.

-       Line 197: One parenthesis seems to be duplicated or missing.

-       Line 213: One parenthesis seems to be duplicated.

-       Line 217: Is "screed" or "screened" the right word?

-       Lines 224-225: It appears that a word is missing in the sentence.

-       Line 240: Replace “Transcriptome Atlas Construction for XinggaoSheep Tissues” by “Transcriptome atlas construction for Xinggao sheep tissues”.

-       Lines 253-254: It appears that the sentence is incomplete.

-       Line 504: Replace “and” by “as well as”.

-       Line 504: Replace “The most expressed genes” by “The study detected the most expressed genes”.

Author Response

Question 1: Abstract section should be 200 words maximum according to “Instructions for Authors”.

Response 1: Thank you for your suggestions. I have compressed the abstract as much as possible, and expressed the central idea of the article. At present, it is 309 words. If it is still a little more, I will further modify it.

The heritability of litter size in sheep is low and controlled by multiple genes, but the research on its related genes is not sufficient. Here, to explore the expression pattern of multi-tissue genes in Chinese native sheep, we selected 10 tissues of the three adult ewes with the highest estimated breeding value in the early study of the prolific Xinggao sheep population. Global gene expression analysis showed that the ovary, uterus, and hypothalamus expressed the most genes;Uniform Manifold Approximation and Projection (UMAP) cluster analysis, these samples were clustered into 8 clusters. Functional enrichment analysis showed that genes expressed in the spleen, uterus, and ovary were significantly enriched in the Ataxia Telangiectasia Mutated Protein (ATM) signaling pathway, and the most genes in the liver, spleen, and ovary were enriched in the immune response pathway. Moreover, we focus on the expression genes of the hypothalamic-pituitary-ovarian axis (HPO) and found that 11 016 genes were co-expressed in the three tissues, and different tissues have different functions, but the oxytocin signaling pathway was widely enriched. To further explore the differences in the expression genes (DEGs) of HPO in different sheep breeds, we downloaded the transcriptome data in the public data, and the analysis of DEGs (Xinggao sheep vs. Sunite sheep in Hypothalamus, Xinggao sheep vs. Sunite sheep in Pituitary, and Xinggao sheep vs. Suffolk sheep in Ovary) revealed that the neuroactive ligand-receptor ect..interactions . In addition, gene subsets of transcription factors (TFs) of DEGs were identified. The results suggest that 51 TFs genes and the homeobox TF may play an important role in transcriptional variation across the HPO. Altogether, our study provided the first fundamental resource to investigate physiological function and regulation mechanisms in sheep. This important data contributes to improving our understanding of the reproductive biology of sheep and isolating effecting molecular markers that can be used for genetic selection in sheep.

Question 2: Introduction section should be written in short paragraphs instead of one very long paragraph.

Response 2: Thank you for your suggestions. I have revised them.

Question 3: In Materials and Methods section, I suggest to include a subsection related to “Gene enrichment or functional analysis”. Also, the website address and the accession date are required for all bioinformatics resources that were used in the study.

Response 3: Thank you for your suggestions. I have revised them in the lasted manuscript.

Question 4: In Discussion section there some very long paragraphs. I suggest separating each of them in 2-3 shorter paragraphs

Response 4: In the manuscript, I have described the discussion section in paragraphs

Question 5: Conclusions section appears to be a brief summary of the results. Instead, I suggest including 2 or 3 conclusive sentences describing the relevance of the study or the importance of the findings. I recommend mentioning litter size, as it was the only reproductive trait involved in this study.

Response 5: I revised this part: In summary, this study aimed to explore the transcriptome atlas and gonad axis of prolific sheep and attempted to elaborate the genetic mechanism to litter size.The study detected the most expressed gens in the hypothalamus and ovary, with the most complex transcript levels. The high-expression genes were mainly involved in the mitochondrial matrix, translation, and ribosome biogenesis. The 51 TF genes may play an important role in HPO. The goal was to lay the foundation for marker-assisted management of sheep breeding and provide important reference data for exploring the genetic basis of fecundity and improving reproductive performance in sheep breeding.

Question 6: In References section, only the first letter of the article title should be capitalized.

Response 6: Thank you for your advice. I check the latest papers published in your magazine, and the articlr titles of the references are all capital letters. I 'm not sure whether to change them. If the revision is revised, I will further amend it.

Question 7: Line 3: I suggest to use “litter size” in the title instead of “fertility”, because it was the only reproductive trait analyzed in the study.

Response 7: Thank you for your suggestions. I have revised them

Question 8: Line 4: Replace “Ovis aries” by “Ovis aries”.

Response 8: The L4 act of the word in the original manuscript

Question 9: Line 20: Replace “The heritability of sheep of litter size is low” by “The heritability of litter size in sheep is low”.

Response 9: Thank you for your suggestions. I have revised them.

Question 10: Lines 26 and 31: I suggest defining UMAP and ATM.

Response 10: Thank you for your suggestions. I have revised them

Question 11: Line 50: I suggest replacing “stock” by “population”.

Response 11: Thank you for your suggestions. I have revised them

Question 12: Line 51: Replace “Sheep have become important model animals in agriculture” by “Sheep have become an important animal model in agriculture”.

Response 12: Thank you for your suggestions. I have revised them

Question 13: Lines 53-54: This sentence appeared to be repeated.

Response 13: Thank you for your suggestions. I have deleted them

Question 14 : Line 65: Replace “The development of” by “The developed”.

Response 14: Thank you for your suggestions. I have revised them

Question 15: Line 72: Separate the round bracket from the text.

Response 15: Thank you for your suggestions. I have revised them

Question 16: Line 87: Replace “secretion” by “production”.

Response 16: Thank you for your suggestions. I have revised them

Question 17: Line 102: Replace “homeostasis” by “regulation”.

Response 17: Thank you for your suggestions. I have revised them

Question 18: Line 106: Please complete the sentence as follows “ewes with high estimated breeding value for litter size…”.

Response 18: Thank you for your suggestions. I have revised them. In this study, we clarified the expression patterns of multiple tissue genes in high-breeding Xinggao sheep, and further comprehensively analyzed the HPO tissues genes and transcription factor. This study aimed to explore the atlas and gonad axis transcriptome and attempted to elaborate the genetic mechanism to further benefit local sheep breeding.

Question 19: Line 111: Replace “Based on my previous” by “Based on previous”.

Response 19: Thank you for your suggestions. I have revised them.

Question 20: Line 112: Separate the square bracket from the text.

Response 20: Thank you for your suggestions. I have revised them.

Question 21: Line 117: Remove the dot.

Response 21: Thank you for your suggestions. I have revised them.

Question22: Line 145: Separate the square bracket from the text.

Response 22: Thank you for your suggestions. I have revised them.

Question 23: Line 148: Replace “ovis ories” by “Ovis aries”.

Rensponse 23: Thank you for your suggestions. I have revised them.

Question 24: Lines 179-181: Please remove blank spaces within the round brackets.

Rensponse 24: Thank you for your suggestions. I have revised them.

Question 25: Line 197: One parenthesis seems to be duplicated or missing.

Rensponse 25: Thank you for your suggestions. I have deleted it.

Question 26: Line 213: One parenthesis seems to be duplicated.

Rensponse 26: Thank you for your suggestions. I have deleted it.

Question 27: Line 217: Is "screed" or "screened" the right word?

Rensponse 27: Thank you for your suggestions. I have revised it.

Question 28: Lines 224-225: It appears that a word is missing in the sentence.

Rensponse 28: Thank you for your suggestions. I have added it.

Question 29: Line 240: Replace “Transcriptome Atlas Construction for XinggaoSheep Tissues” by “Transcriptome atlas construction for Xinggao sheep tissues”.

Rensponse 29: Thank you for your suggestions. I have revised it.

Question 30: Lines 253-254: It appears that the sentence is incomplete.

Rensponse 30: Thank you for your suggestions. I have revised it. Cluster analysis of the top 10 genes expression in each cluster, the same cluster of genes, and may have similar functions, from the same type of tissue.

Question31: Line 504: Replace “and” by “as well as”.

Rensponse 31: Thank you for your suggestions. I have revised it.

Question 32: Line 504: Replace “The most expressed genes” by “The study detected the most expressed genes”.

Resnsponse 32: Thank you for your suggestions. I have revised it.

Round 2

Reviewer 1 Report

line 113-116 the sentence is still incorrect

line 377 line 552 and elsewhere the term "gonadal axis " is incorrect

what was the exact age of xing bao sheep - if they were used for " between breeds " comparison it is important. The age between Suffolk and Sunite breed was different. Age can be a source of differential expression and interfer with the results.

Figure 3 in unreadable

Author Response

Dear Reviewer,

Thank you for your valuable comments for my manuscript during your busy schedule. Here, please allow me to express my best wishes to you! I wish you good health and success in your work!

I have answered all your questions and suggestions about the manuscript in below, and made revisions using the " Reply to Reviewers" function in the manuscript. I have submitted it to you with this reply for your criticism and correction. If there is still something unreasonable, please let me know in time. I will look forward to it very much.

Thank you!

Kind regards!

Point 1: line 113-116 the sentence is still incorrect

Response 1: The revisions has been completed in accordance with the comments.

In this study, we clarified a comprehensive transcriptome analysis of multiple tis-sues in sheep. Our results uncovered the transcriptome atlas and analyzed gene ex-pression as well as transcription factors in HPO tissue. It was also demonstrated that different tissues exerted a diverse range of physiological functions, including behavior, immune response, and development. Finally, we have successfully identified the gene subsets of TFs and discussed the potential mechanism in the HPO axis. Overall, our findings offer a comprehensive analysis of gene expression variation and physiological function specialization in multiple tissues of high-fecundity sheep. It provides basic and useful resources for further study of molecular characteristics and transcriptional regulation of sheep.

Point 2: line 377 line 552 and elsewhere the term "gonadal axis " is incorrect

Response 2: The revisions has been completed in accordance with the comments.

I modified the “gonadal axis” in the manuscript to HPO axis.

Point 3: what was the exact age of xing bao sheep - if they were used for " between breeds " comparison it is important. The age between Suffolk and Sunite breed was different. Age can be a source of differential expression and interfer with the results.

Response 3: The revisions has been completed in accordance with the comments.

three 3.5-year-old adult ewes with similar ages and weights were selected with higher estimated breeding values (0.32, 0.29, 0.21).

Point 4: Figure 3 in unreadable

Response 4: The revisions has been completed in accordance with the comments. Figure 3 I have re-uploaded
